# Comparing Even with Skewed Dietary Protein Distribution Shows No Difference in Muscle Protein Synthesis or Amino Acid Utilization in Healthy Older Individuals: A Randomized Controlled Trial

**DOI:** 10.3390/nu14214442

**Published:** 2022-10-22

**Authors:** Thomas Ehlig Hjermind Justesen, Simon Elmer Jespersen, Thomas Tagmose Thomsen, Lars Holm, Gerrit van Hall, Jakob Agergaard

**Affiliations:** 1Institute of Sports Medicine Copenhagen, Department of Orthopedic Surgery, Copenhagen University Hospital—Bispebjerg and Frederiksberg, 2400 Copenhagen NV, Denmark; 2Center for Healthy Ageing, Department of Clinical Medicine, University of Copenhagen, 2200 Copenhagen N, Denmark; 3School of Sport, Exercise and Rehabilitation Sciences, University of Birmingham, Birmingham B15 2TT, UK; 4Clinical Metabolomics Core Facility, Rigshospitalet, 2200 Copenhagen N, Denmark

**Keywords:** dietary protein, skeletal muscle, aging, protein distribution, muscle protein synthesis

## Abstract

Sarcopenia is a multifactorial disease that limits autonomy for the growing elderly population. An optimal amount of dietary protein has shown to be important to maintain muscle mass during aging. Yet, the optimal distribution of that dietary protein has not been fully clarified. The aim of the present study was to examine whether an even, compared to a skewed, distribution of daily dietary protein leads to higher muscle protein synthesis and amino acid utilization. Twelve healthy males and twelve healthy females aged between 65 and 80 years were block randomized to either an even (EVEN, *n* = 12) or skewed (SKEWED, *n* = 12) dietary protein distribution for three daily main meals. Seven days of habituation were followed by three trial days, which were initiated by oral intake of deuterium oxide (D_2_O). The dietary protein throughout all trial meals was intrinsically labelled with ^2^H_5_-phenylalanine. Blood samples were drawn daily, and muscle biopsies were taken before and at the end of the trial to measure muscle protein synthesis (FSR) and muscle protein incorporation of the dietary-protein-derived tracer. Muscle protein FSR was no different between the two groups (EVEN 2.16 ± 0.13%/day and SKEWED 2.23 ± 0.09%/day, *p* = 0.647), and the muscle protein incorporation of the intrinsically labeled ^2^H_5_-phenylalanine tracer was not different between the two groups (EVEN 0.0049 ± 0.0004 MPE% and SKEWED 0.0054 ± 0.0003 MPE%, *p* = 0.306). In conclusion, the daily distribution pattern of the dietary protein did not affect muscle protein synthesis or the utilization of dietary protein.

## 1. Introduction

The degenerative loss of skeletal muscle mass and function, also known as sarcopenia, is a factor with great negative influence on the physical health and autonomy of older adults [1]. Besides being important for the ability to move around and carry out everyday tasks, skeletal muscle is also important for our metabolism [2]. In men >60 years of age, muscle strength is a predictor of all-cause mortality [3]; muscle mass is a predictor of all-cause mortality in women >60 years [4], and low muscle strength with low physical performance has been shown to be a predictor of mortality independent of other mortality risk factors such as ischemic heart disease, activities of daily living, age, or gender [5]. Hence, delaying or minimizing the development of sarcopenia is a target of efforts to maintain the health and autonomy of the aging population.

The etiology behind the development of sarcopenia is multifactorial, and is the result of both age-dependent cellular changes as well as age-related changes in lifestyle, such as diet and physical activity [6,7,8]. Regarding diet, the recommended daily protein intake for elderly people is debated; however, consensus is arising that people >65 years of age should have a daily protein intake of 1.2 g per kg body weight to maintain skeletal muscle mass [9]. 

The net protein balance of skeletal muscle is determined by protein synthesis and protein breakdown, and the balance between these two turnover rates changes throughout the day as a response to anabolic stimuli such as protein-feeding and resistance exercise [10]. Whereas studies have tried to clarify how much protein is needed on daily basis, there has been less focus on ways to optimize daily distribution of the protein to improve utilization of the ingested protein and in order to ensure an optimal net balance in skeletal muscle protein turnover in the elderly. Acute metabolic studies with the application of infused amino acid tracers have shown a graded increase in muscle protein synthesis (MPS) in the elderly with increasing doses of protein or essential amino acid (EAA) intake [8,11,12]. A protein content above 0.40 g per kg body weight (BW) has shown to maximally stimulate postprandial MPS in older men [13]. This suggests that to optimize MPS at each meal during the day and positively affecting the skeletal muscle net protein balance, elderly individuals could benefit from ingesting 0.40 g per kg BW of protein in all main meals during the day, i.e., an even protein distribution. 

Nevertheless, it seems that most elderly people have a skewed protein intake during the day, where most of the daily protein is ingested at dinner while less protein is ingested at breakfast and lunch [14,15,16]. With a skewed protein distribution, breakfast and lunch meals could have a protein content below the 0.40 g per kg BW needed to maximally stimulate MPS [16]. In addition, it is possible that more protein than needed could be consumed at dinner, with the excess amino acids being oxidized and wasted. Consequently, the skeletal muscle’s net protein balance could be stimulated less optimally during a day with a skewed protein distribution when compared to evening out the protein distribution and, thereby, having multiple meals with a protein intake reaching the 0.4 g per kg BW.

It has been indicated that evenness of dietary protein distribution is associated with a higher muscle mass [17]. In addition, the importance of daily protein distribution has been shown in a cross-sectional study where men and women aged 75–96 with a skewed protein distribution had increased risk of frailty [18]. In contrast, in elderly malnourished at-risk patients in an inpatient rehabilitation unit, a skewed protein distribution had a positive effect on lean body mass compared to an even distribution [19]. When looking at acute metabolic studies, the effect of the protein distribution is unclear. Two studies by Kim and colleagues found no effect of an even protein distribution compared to a skewed distribution [20,21]. Yet, in adults aged 25–52, Mamerow and colleagues showed a greater mixed-muscle fractional synthesis rate with an even protein distribution compared to a skewed protein distribution [22].

Hence, the current literature is sparse and ambiguous. Therefore, the aim of the present study was to assess the importance of daily protein distribution through an integrated measurement of muscle protein synthesis and amino acid utilization over consecutive days. Importantly, the study was designed as a randomized controlled trial that mimics everyday life, looks at a demographic relevant population, and controls sufficiently for total protein intake, protein distribution, and total calorie intake. It was hypothesized that an even vs. a skewed daily protein distribution would result in a greater utilization of dietary amino acids for *de novo* muscle protein synthesis.

## 2. Materials and Methods

### 2.1. Subjects

Before inclusion in the study, each subject had the study design, purpose, and possible risks explained to them. Subsequently, all subjects gave their written consent to participate in the protocol, which adhered to the Declaration of Helsinki and was approved by the Ethics Committee of Copenhagen and Frederiksberg (H-18026529). The study is registered at ClinicalTrials.net (NCT03870425).

*Inclusion protocol*. Twelve healthy elderly males and twelve healthy elderly females were block randomized to either the EVEN (*n* = 12) or the SKEWED (*n* = 12) protein distribution throughout all 11 days (Table 1). See Figure 1 for CONSORT flow diagram.

Exclusion criteria for this study were an age below 65 or above 80 years of age, body mass index (BMI) below 18.5 or above 30 kg/m^2^, smoking, vegetarianism, diabetes or any other metabolic diseases, gastrointestinal diseases, impaired kidney or liver function, inflammatory diseases, hypertension, signs of arteriosclerosis, or more than 5 h of weekly systematic training, except for activity associated with transportation as well as dancing and stretching. 

Written informed consent was obtained from all subjects before enrollment to the study. Subject’s height, weight and blood pressure were measured, and blood samples were taken for health screening. In addition, DEXA scans were carried out to determine lean body mass (LBM).

### 2.2. Diet

*Diet allocation.* Before each trial, subjects were randomly allocated to either an even or a skewed protein distribution for the entire trial, which consisted of a 7-day habituation period followed by a 3-day trial period at the hospital (Figure 2). Subjects were block randomized in three blocks with 8 subjects in each block in order to ensure an equal number of subjects as well as an equal gender distribution in the EVEN and SKEWED groups.

The food ingested by the subjects during the entire trial was made at the Research Kitchen at the Department of Human Nutrition, University of Copenhagen. 

*Habituation period*. Prior to the 3 trial days at the hospital, subjects went through 7 days of dietary habituation (Figure 2) where they maintained their everyday lifestyle but were served a standardized diet for all 7 days, with either an even or skewed protein distribution. The standardized meals (breakfast, lunch, and dinner) were prepacked for all seven days. Daily calorie intake was determined by the Harris–Benedict equation, calculated on the basis of total LBM and with an activity factor of 1.53 [23]. The calorie intake during habituation was divided into four levels. Subjects with LBM between 30–38 kg received 7.7 mega joule (MJ) daily, subjects with LBM between 39–46 kg received 8.8 MJ daily, subjects with LBM between 47–54 kg received 9.9 MJ daily, and subjects with LBM between 55–63 kg received 11 MJ daily. The calories of the main meals were equal, with 33% of the total daily calorie intake at breakfast, 33% at lunch, and 33% at dinner. In addition, subjects were allowed to choose a snack low in protein twice daily. The subjects were instructed on which type of snacks were allowed (low in protein) and not allowed (high in protein). Additionally, the subjects filled out a questionnaire each day answering whether they had eaten a snack that day, and if so, what snack had been eaten. The reporting of the snacks through the questionnaire was continuously monitored by the researchers. 

The daily protein intake during habituation was between 1.3–1.6 g/kg LBM in both groups, within all four levels of calorie intake: 7.7, 8.8, 9.9, and 11 MJ, respectively. Importantly, the distribution of the daily protein intake was 33% at breakfast, 33% at lunch, and 33% at dinner for the EVEN group, whereas in the SKEWED group, the distribution was 17% at breakfast, 17% at lunch, and 66% at dinner.

Subjects were thoroughly instructed to remain compliant with their diets and, except for the snacks, not to ingest anything containing energy at the meals. Tap water, sparkling water without energy, and coffee/tea without milk or sugar were allowed. Subjects had to eat breakfast between 7.00–9.00 a.m., lunch between 11.00 a.m.–1.00 p.m., and dinner between 6.00–8.00 p.m. Importantly, the subjects were instructed not to skip any meals, not to save food for later, to finish their meals, and to try an extra time if the meal was difficult to finish. However, if they were unable to finish a meal, they were instructed to freeze the leftovers and bring them to the researchers during the following trial days at the hospital. Here, the leftovers would be measured and registered. Subjects were allowed to add salt, pepper, or other seasonings to the meals.

*Trial days*. During the trial days, the majority of the dietary protein in all meals and snacks consisted of minced beef intrinsically labeled with ^2^H_5_-phenylalanine in order to measure the utilization of the dietary protein. To limit the dietary dilution of the ^2^H_5_-phenylalanine tracer from the minced meat with phenylalanine from other protein sources, whole foods very low in protein were chosen for the remainder of the meals and snacks. Information on the production of intrinsically labeled minced meat has previously been published [24]. The only meal not containing intrinsically ^2^H_5_-phenylalanine-labeled minced meat was the initial breakfast on the first trial day, as this meal was consumed before the background blood sample was collected. 

During the three trial days at the hospital, subjects continued eating according to their respective protein distribution with a total protein intake of 1.5 g/kg LBM/day, equivalent to approximately 1.0–1.2 g/kg BW, in both the EVEN and SKEWED group (Table 2). The protein intake of the EVEN group was divided into 0.450 g/kg LBM at breakfast, 0.075 g/kg LBM as a noon snack, 0.450 g/kg LBM at lunch, 0.075 g/kg LBM as an afternoon snack, and 0.450 g/kg LBM at dinner (30%-5%-30%-5%-30% distribution). The protein intake of the SKEWED group was divided into 0.225 g/kg LBM at breakfast, 0.075 g/kg LBM as a noon snack, 0.225 g/kg LBM at lunch, 0.075 g/kg LBM as an afternoon snack, and 0.900 g/kg LBM at dinner (15%-5%-15%-5%-60% distribution). With this protein distribution design, the present study is comparable to previous studies by Mamerow and colleagues and Kim and colleagues [20,22]. The total daily intake of calories was calculated on the basis of the individual LBM of each subject with the Harris–Benedict equation, with a correction factor of 1.53 for light activity. All meals were prepared and weighted by the researchers. The subjects were instructed to finish all meals, and this was controlled by the researchers. Tap water, sparkling water without energy, and coffee/tea without milk or sugar were allowed. Breakfast was served at 8.00 a.m., noon snack at 10.15 a.m., lunch at 12.30 p.m., afternoon snack at 3.00 p.m., and dinner at 5.45 p.m. For both groups, the total daily calories were divided as follows: 30% in the morning, 5% as a noon snack, 30% at lunch, 5% as an afternoon snack and 30% at dinner. Each day at 6.00 p.m., subjects were allowed to return home. They would then return to the hospital each day at 7.45 a.m. Subjects were instructed not to ingest anything containing energy or protein while at home. Only tap water, sparkling water without energy, and coffee/tea without milk or sugar were allowed while subjects were not at the hospital.

### 2.3. Activity and Activity Monitoring

*Activity monitoring*. During both the habituation period and the trial days at the hospital, the subjects were fitted with an ActivPal activity tracker (PAL Technologies Ltd., Scotland, UK). Activity was tracked for 3 out of 7 days during the habituation period and during the 3-day trial period. The ActivPal would collect data from the first day at 6.00 a.m. until the fourth day at 6.00 a.m. During the trial days at the hospital, the ActivPal would collect data from the first day at 11.00 a.m. until the fourth day at 6.00 a.m. The ActivPal was taped to the middle of the anterior side of the subject’s thigh. The ActivPal measured time spent lying, sitting, and standing during the day, while also registering daily step count. Two ActivPals failed to collect data during the habituation period, one in the SKEWED group and one in the EVEN group.

*Activity*. Throughout the habituation days, the participants were asked to maintain their usual activity levels. During the trial days at the hospital, the activity level was controlled by the researchers. 

During the three initial trial days, subjects had three planned daily activities. All activities were supervised by a researcher. At 10.30 a.m., a walk on a predetermined 3 km route in the vicinity of the hospital was completed, lasting for approximately 45 min. At 2.00 p.m., 15 min of cycling was performed on a cycle ergometer with a cadence of 60–80 rpm and fitted with a heart rate monitor to ensure exercise at 60–80% of their estimated maximal heart rate. 

Finally, at 4.30 p.m., an approximately 30 min walk was completed along a predetermined 1.5 km route around the hospital that included climbing of 357 stairs at an easy walking pace. On the first trial day, the 3 km walk was skipped in order to collect the initial background muscle biopsy and blood sample.

### 2.4. Measurements

*Blood sampling*. On the first trial day, a background blood sample was collected at approximately 9.00 a.m. before consumption of intrinsically labeled ^2^H_5_-phenylalanine beef and oral D_2_O intake. For the remainder of the trial days, blood samples were taken at 9.00 a.m. Blood samples were collected in EDTA tubes. After 10–30 min on ice, the tubes were centrifuged at 3200× *g* for 10 min at 4 °C, after which the plasma were aliquoted and stored at −80 °C until further analysis.

*Muscle biopsies*. At approximately 10.00 a.m. on the first trial day a background muscle biopsy was taken, and a second biopsy was taken on the fourth trial day at 10.00 a.m.

Initially, by randomization, either the dominant or the non-dominant leg would be chosen as the site for all biopsies for each subject. Biopsies were taken with 3 cm between each site, and the order of the biopsies within the leg was randomized as well. 

The muscle biopsies were taken from the lateral portion of the vastus lateralis muscle in the leg under local anesthetic treatment (lidocaine, 1%). The biopsies were taken with 4 mm Bergström biopsy needles (Stille, Stockholm, Sweden) with manual suction. From the muscle specimen, blood, visible fat, and connective tissue were quickly removed, and the muscle specimen was rinsed with saline water before being snap frozen in liquid nitrogen and stored at –80 °C until further analysis.

*D_2_O administration*. Administration of D_2_O (DLM-2259-PK, Cambridge Isotope Laboratories Inc., Tewksbury, MA, USA) was used to measure the muscle protein synthesis during the trial days. The 99.8% D_2_O, diluted 1:1 in tap water, was administered orally in six hourly boluses during the first trial day starting at 11.00 a.m. to minimize the adverse effects of D_2_O [25]. In total, the subjects received 5.25 mL 99.8% D_2_O/kg LBM, and all subjects consumed the entire dose.

### 2.5. Mass Spectrometry Analysis

For the mass spectrometry analysis of ^2^H-alanine and ^2^H_5_-phenylalanine muscle enrichment, 20 mg of muscle tissue was used. The muscle tissue was homogenized in homogenization buffer (0.02 M Tris [pH 7.4], 0.15 M NaCl, 2 mM EDTA, 2 mM EGTA, 0.5% TritonX-100 and 0.25 M sucrose). The proteins were hydrolyzed by addition of 1 mL 1 M HCl and 1 mL resin slurry, and left overnight at 110 °C.

For mass spectrometry analysis of ^2^H-alanine precursor enrichment from the plasma samples, 200 μL of plasma per sample was used. Isotopically labeled internal standards (uniformly labeled 13C/15N) of each amino acid were added to the plasma in a 50% acetic acid solution.

Hereafter, the solution from either the muscle or plasma samples was poured over cation exchange columns with resin (AG 50 W-X8 resin, Bio-Rad Laboratories, Hercules, CA, USA) which had been prepared by adding 3 × 2 mL 1 M HCl. The resin columns were washed 5 times with 3 mL deionized water before the amino acids were eluted into collection vials by adding 2 × 2 mL 4 M NH_4_OH. The solution of each of the muscle samples was divided into two portions for liquid chromatography–tandem mass spectrometry (LC–MS/MS) and gas chromatography–combustion–isotope ratio mass spectrometer (GC–C–IRMS) analysis, respectively.

To measure the plasma ^2^H-alanine precursor enrichment and muscle incorporation of the dietary-protein-derived ^2^H_5_-phenylalanine, the solvents eluted from the resin columns were evaporated under a stream of N_2_ flow at 70 °C and samples were derivatized into their PITC derivative with phenylisothiocyanate (PITC). Ten microliters of the derivatized samples were loaded and analyzed by LC–MS/MS (TSQ Quantiva; Thermo Fisher Scientific, San Jose, CA, USA) as described elsewhere [26].

For determining the ^2^H-alanine abundance in the muscle tissue samples, amino acids in the solvent eluted from the resin column were converted to the N-acetyl-propyl (NAP) derivatives and analyzed by GC–C–IRMS as previously described by Bornø et al. [27]

### 2.6. Fractional Synthesis Rate Calculation

The fractional synthesis rate (FSR) of muscle proteins was calculated form the direct incorporation rate,
$$\mathrm{FSR}=\left[\frac{∆E_{protein}}{{\hat{E}}_{precusor}\times ∆t}\right]\times 100$$

The FSR is expressed in % × day^−1^ and was calculated from the change in muscle protein tracer enrichment *(*Δ*E_protein_*) from the background muscle tissue sample taken on the first trial day to the muscle tissue sample obtained on the last trial day, divided by the weighted average of the precursor enrichment measured from the plasma of the daily blood sampling (*Ê_precursor_*) multiplied by the incorporation time (Δ*t*).

### 2.7. Statistics

All data were analyzed with the Shapiro–Wilk normality test before parametric statistics were applied. Subject characteristics, total calorie intake, muscle protein synthesis, and ^2^H_5_-phenylalanine incorporation were analyzed by two-tailed unpaired *t*-test. Data on protein and calorie intake per meal during the 7 habituation days were analyzed by two-way ANOVA with repeated measures for meal (breakfast, lunch, dinner). Activity data was analyzed by two-way ANOVA with repeated measures for time (habituation vs. trial period). Subject characteristics and protein and calorie intake are presented as mean ± SD. The remaining data are presented as mean ± SEM with individual data indicated. All data was analyzed by Sigma Plot version 13.0 (Systat Software Inc., San Jose, CA, USA).

## 3. Result

### 3.1. Subject Characteristics

Before the trial, no significant differences existed between groups in regard to age (*p* = 0.784), height (*p* = 0.648), weight (*p* = 0.132), BMI (*p* = 0.163), LBM (*p* = 0.502), fat percentage (*p* = 0.363), systolic blood pressure (BP) (*p* = 0.439), diastolic BP (*p* = 0.480), HbA1c (*p* = 0.702), thyroid-stimulating hormone (TSH) (*p* = 0.605), total cholesterol (*p* = 0.557), high-density lipoprotein (HDL) (*p* = 0.081), or low-density lipoprotein (LDL) (*p* = 0.940) (Table 1).

### 3.2. Diet

During the habituation period, no differences in protein intake per kg LBM were seen between meals in the EVEN group, whereas the intake was greater at dinner compared to breakfast (*p* < 0.001) and lunch (*p* < 0.001) in the SKEWED group (Figure 3A). The protein intake per kg LBM was greater at breakfast (*p* < 0.001) and lunch (*p* < 0.001) in EVEN vs. SKEWED, whereas the intake at dinner was smaller in EVEN vs. SKEWED (*p* < 0.001). No significant differences between the two groups or between meals were found for total kcal per day (Figure 3B). The total daily calorie intake was the same for the EVEN and SKEWED groups (Figure 3C).

### 3.3. Activity and Activity Monitoring

No significant main effect of intervention was found for any of the five activity parameters (Table 3). A significant effect of time was found, with a greater number of steps (*p* < 0.001) and step time (*p* = 0.005) during the trial period compared to habituation and a shorter stand time (*p* < 0.001) during the trial period compared to habituation. The energy expenditure (MET) was slightly lower during the trial period compared to habituation (*p* = 0.006). No significant effect of time was found for sit-lie time (*p* = 0.084)

### 3.4. Muscle Protein Synthesis and Intrinsically Labeled Tracer Incorporation

During the trial period, no significant difference between the two groups was found for mixed-muscle protein FSR (*p* = 0.647) (Figure 4A). No significant difference between the two groups was found for incorporation of dietary-protein-derived ^2^H_5_-phenylalanine (*p* = 0.387) (Figure 4B).

## 4. Discussion

The present study explored the importance of how the daily protein intake is distributed with regards to muscle protein synthesis and utilization of dietary protein. To the knowledge of the authors, this is the first study on healthy older adults to explore the effect of dietary protein distribution in an energy-balanced diet by applying an integrated measurement of muscle protein synthesis over consecutive days and assessing the utilization of dietary-protein-derived amino acids using intrinsically labeled protein. It was found that a sufficient daily protein intake at 1.5 g protein per kg LBM with either an even or skewed distribution resulted in the same rates of muscle protein synthesis, as well as similar utilization of the dietary-protein-derived amino acids for synthesis of muscle proteins. The study was performed with a randomized controlled trial design that mimicked everyday life, included a demographically relevant population, employed strict dietary control that ensured a distinct protein distribution in the two intervention groups—EVEN and SKEWED, respectively—and, at the same time, ensured identical total protein intake and total calorie intake for both groups. 

The integrated measurement of muscle protein synthesis over multiple days is, to a large degree, affected by physical activity level and diet habits [28]. Therefore, the translatability of the muscle protein synthesis outcome in a parallel group design, as in the present study, is affected by the ability to control for physical activity level, energy expenditure, and dietary intake. To ensure that the prior diet habits would not influence the change to an EVEN or SKEWED distribution, a run-in habituation period of 7 days was performed before any outcome measurements were commenced. Furthermore, as previously described, the diet was designed based on the subject’s LBM, and the protein and calorie intake were controlled through daily reporting during the habituation period as well as by food preparations and observations by the researchers during the trial period. The activity levels of the subjects were monitored both during the habituation days and during the trial days in the two groups. Notably, the physical activity for the subjects during the trial was not designed to be physical training, but merely to ensure an activity level and energy expenditure comparable to the habitual activity as observed during the habituation period. Despite differences being seen in stand time, step time, and total steps per day from habituation to the trial period, only a slight difference in the energy expenditure was seen. Importantly, no differences in the monitored physical activity were observed between the EVEN and SKEWED groups, which otherwise could have affected the integrated measurement of muscle protein synthesis. Therefore, by strict dietary and activity design and control, it is possible to translate the outcome of the study to be based on dietary protein distribution—EVEN and SKEWED, respectively—as the major diverging parameter.

The results from previous studies exploring the muscle protein synthesis response from an even or skewed dietary protein distribution are inconsistent. Studies from Kim and colleagues showed no difference in mixed-muscle FSR with an even or skewed dietary protein intake [20,21], which is in line with the findings of the present study. In contrast, the findings of Mamerow et al. [22] contradict the results of these studies, showing a greater mixed-muscle FSR with an even compared to a skewed protein distribution. The protein dose provided in the study by Kim et al. was given at a low dose of 0.8 g per kg BW per day or high dose of 1.5 g per kg BW per day, and in the study by Mamerow et al., protein dose was 1.2 g per kg BW; in both studies, the dietary protein was given in mixed meals of macronutrients. Therefore, the protein dose and source of the meals was comparable to the present study. It should be noted that the study by Mamerow and colleagues is on a younger population compared to the present study and the studies by Kim and colleagues. Whether an age difference exists in the effect of an even dietary protein distribution can only be speculated. There could be several mechanisms for such a distinct effect, e.g., different protein metabolism, or higher habitual activity level and, thereby, greater amino acid demand in younger vs. older adults. In the above-mentioned studies by Kim et al. and Mamerow et al., muscle protein synthesis was measured over 24 h with a stable-isotope amino acid tracer infusion. Except for a 15–30 min treadmill walk during these 24 h, the tracer infusion necessitated that the subjects be primarily bed-ridden during the 24 h tracer infusion. In such a setting, the stimulatory response of habitual daily activity on muscle protein synthesis becomes negligible. In contrast, the subjects in the present study were not restricted to bed rest, as the tracer was given orally on day 1 which warranted a subsequent synthesis measurement under free-living conditions. Furthermore, the integrated measurement over multiple days decreased the risk of interference by day-to-day variations in protein synthesis. 

In addition to the measurement of muscle protein synthesis, the present study assessed the muscle utilization of the ingested dietary protein by consuming intrinsically amino-acid-labeled meat at all meals during the three-day trial period. As mentioned in the introduction, it has been shown that a protein intake of 0.4 g per kg body weight per meal is needed to maximally stimulate muscle protein synthesis in older adults [13], which formed the basis of the meal-size design in the present study. Therefore, it was hypothesized that the greater protein intake at the evening meal of the SKEWED distribution (0.9 g per kg LBM) would not be fully utilized, and an excess appearance of amino acids into circulation would be oxidized instead of being used for *de novo* synthesis of proteins. It should be noted that the study indicating a maximum stimulation of muscle protein synthesis at 0.4 g per kg body weight was performed with a bolus intake of pure protein, and not together with a mixed meal. The amino acid absorption after a bolus of protein is fast and transient, whereas a more prolonged release of amino acids into circulation is seen after intake of protein in a mixed meal [24,29]. Such a prolonged release could potentiate full utilization of a greater protein intake over the post-prandial period, which could be why a difference was not seen between the EVEN and SKEWED distribution groups in muscle-tissue incorporation of the intrinsically labeled ^2^H_5_-phenylalanine tracer in the current study. It should be noted that, when exploring the incorporation of a dietary derived amino acid tracer over multiple days, a potential recirculation of the tracer could occur due to an efflux of the tracer from body proteins because of protein breakdown. However, by limiting the study period to three days, the recirculation is minor and, at the same time, the study duration makes it possible still to measure the muscle incorporation of the dietary derived amino acid tracer.

The study was conducted with a total daily protein intake that was sufficient. The daily protein intake during the trial period, 1.5 g of protein per kg LBM, is equivalent to approximately 1.0–1.2 g of protein per kg body weight. Furthermore, the activity level of the subjects was low to moderate. Thus, the conclusion that no difference between even and skewed protein intake exists should not be extrapolated to either a situation where the total protein intake is substantially lower or a situation with highly physically active older adults. In both such situations, the metabolic amino acid demand would be greater and a potential effect of distributing the dietary protein throughout the day could be significant. Previously, we have indicated that an even protein intake is associated with a higher muscle mass [17], and it has been shown that a skewed protein intake was associated with a higher risk of frailty [18]. Therefore, the dietary protein distribution could be clinically relevant, e.g., in patients at risk of sarcopenia that could also experience malnourishment due to a reduced appetite. Thus, as previously discussed, in a practical setting, a dietary protein intake that is evenly distributed throughout the daily meals could increase the total daily protein intake and, thereby, ensure a sufficient protein intake to maintain the protein pool of the body [17].

Although the diet of the subjects was controlled to the best of our abilities, the subjects were at home during the entire seven-day habituation period, and during the three trial days, the subjects were at home between 6:00 p.m. and 8:00 a.m. each day. Therefore, the dietary control is limited by the inability to closely observe the subject during the habituation period and at nighttime during the trial period. Nevertheless, with clear instructions to the subjects throughout the entire experiment on the importance of complying with the dietary intervention and with registration of all meals and snacks, it is believed that compliance with the diet was high and deviations were negligible.

## 5. Conclusions

Through a randomized controlled trial on healthy older adults with a sufficient total daily dietary protein intake distributed evenly or skewed throughout daily meals, no difference in the level of muscle protein synthesis was found. Furthermore, no difference in the utilization of the dietary protein in the skeletal muscle was seen between the EVEN and a SKEWED protein distributions.

## Figures and Tables

**Figure 1 nutrients-14-04442-f001:**
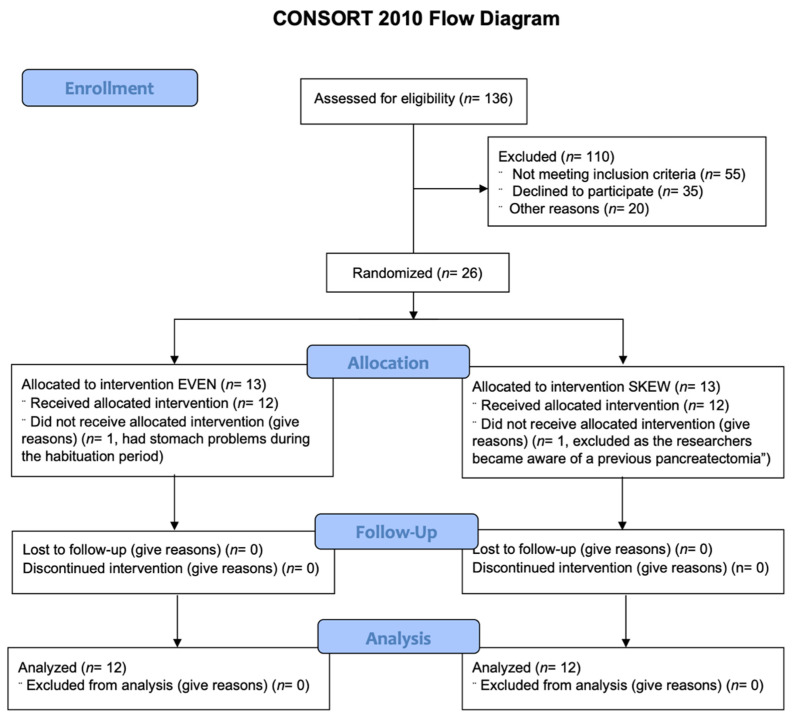
CONSORT diagram. 136 potential subjects were assessed for eligibility. 110 were excluded either due to not meeting inclusion criteria (55), declining to participate (35) or other reasons (20), leaving 26 subjects fit for inclusion. The 26 remaining subjects were randomized to either the EVEN or SKEWED intervention group. 1 subject from each intervention group were lost during the habituation period, leaving 12 subjects in each intervention group that all completed the entire study and who were all included in the later analysis.

**Figure 2 nutrients-14-04442-f002:**
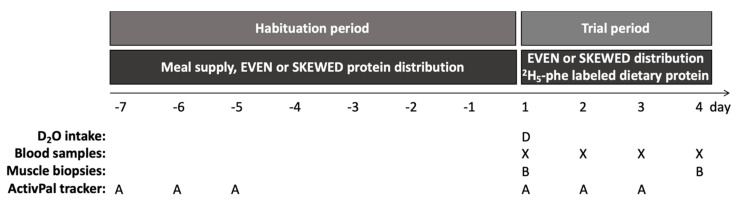
Study design. Indicating the initial 7-day dietary habituation period from day -7 to 1, with habituation to the skewed or even protein distribution by home delivered meals, followed by the three-day trial period started on day 1. Subjects showed up at the hospital from 8.00 a.m. to 18.00 p.m. on all trial days. After background muscle biopsy (B) and blood sample (X) were taken on day 1, the subjects ingested 5.25 mL D_2_O/kg LBM and the trial meals were commenced. All trial meals were prepared by the researchers and contained intrinsically labeled ^2^H_5_-phenylalanine protein, and the meal intake was monitored to ensure full compliance. On day 2 and 3, subjects had a blood sample taken and followed the schedule of the trial meals. On day 4 subjects had breakfast, and two hours later muscle biopsy and blood sample were collected. During the habituation period, the subjects followed their habitual lifestyle except from adhering to the meals, whereas during the trial days a pre-determined activity schedule (walks and cycling) was followed. The activity level was monitored (A) during the initial 3 days during the habituation and trial period, respectively.

**Figure 3 nutrients-14-04442-f003:**
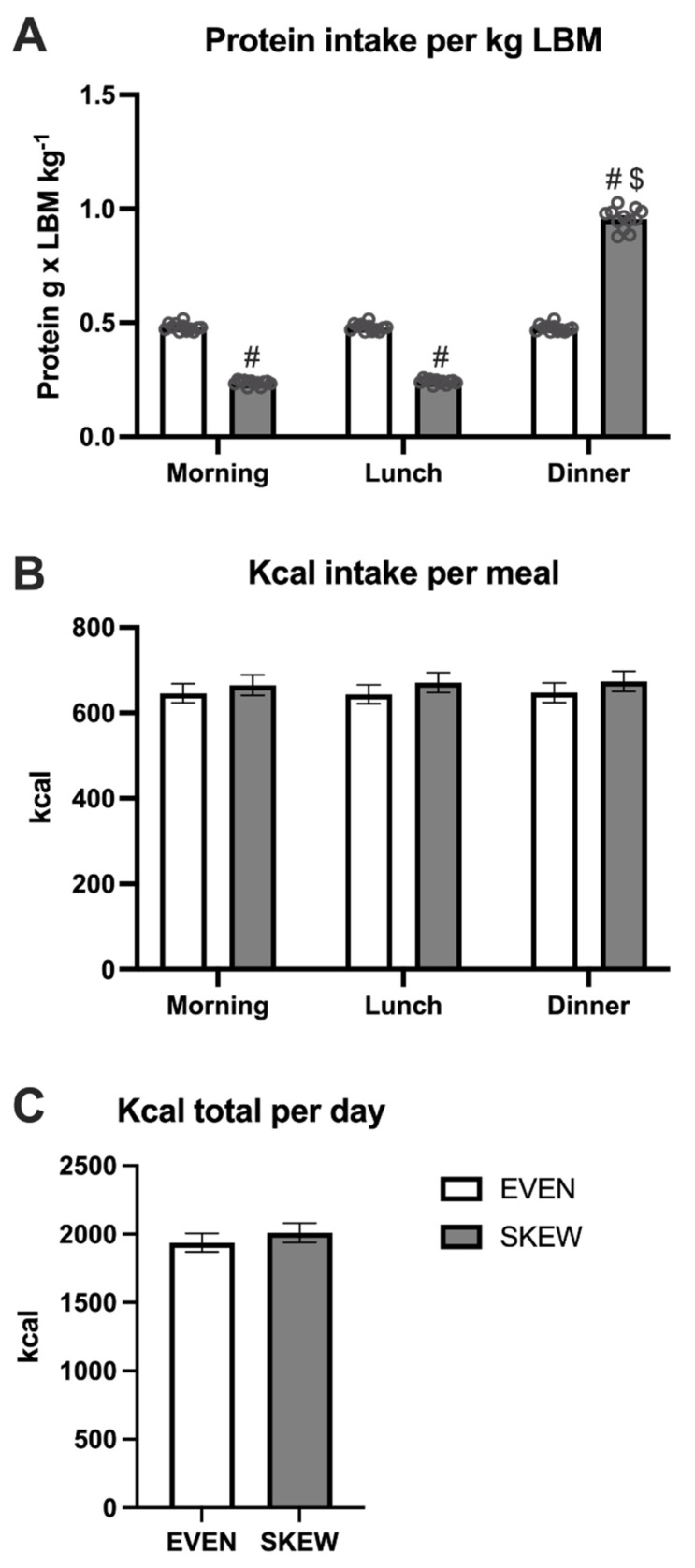
The daily intake of (**A**) protein per meal, (**B**) kcal per meal, and (**C**) total ingested kcal during the habituation period. 3A: No differences in protein intake per kg LBM were seen between meals in EVEN, whereas the intake was greater at dinner compared to breakfast (*p* < 0.001) and lunch (*p* < 0.001) in SKEWED. The protein intake per kg LBM was greater at breakfast (*p* < 0.001) and lunch (*p* < 0.001) in EVEN vs. SKEWED, whereas the intake at dinner were smaller at EVEN vs. SKEWED (*p* < 0.001). 3B: No significant differences between the two groups or between meals were found for total kcal per day. 3C: The total daily calorie intake was the same for the EVEN and SKEWED group. Data shown as mean ± SEM. # denote different from EVEN, $ denote different from breakfast and lunch.

**Figure 4 nutrients-14-04442-f004:**
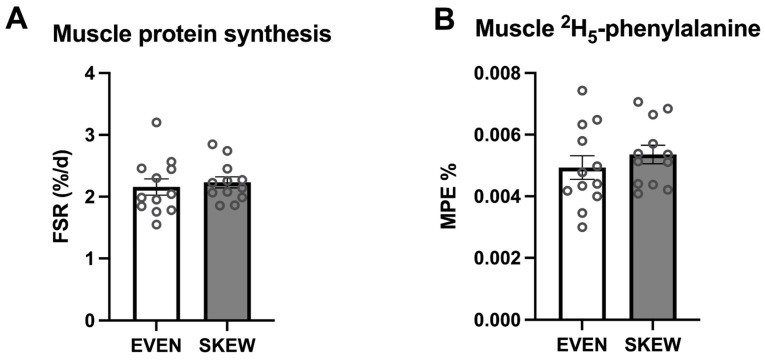
(**A**) Muscle protein FSR and (**B**) muscle ^2^H_5_-phenylalanine incorporation. 4A: No significant difference between the two groups was found for mixed muscle protein FSR (*p* = 0.647). 4B: No significant difference between the two groups was found for incorporation of the dietary protein derived ^2^H_5_-phenylalanine (*p* = 0.387). Data shown as mean ± SEM, with individual data indicated as circles.

**Table 1 nutrients-14-04442-t001:** Subject characteristics at inclusion.

	EVEN	SKEWED	*p*-Value
N	12	12	*-*
Sex Ratio (w/m)	6/6	6/6	-
Age (years)	69 ± 4	70 ± 3	*0.784*
Height (cm)	169.1 ± 9.5	171.0 ± 8.5	*0.648*
Weight (kg)	72.4 ± 11.6	79.0 ± 8.7	*0.132*
BMI (kg/m^2^)	25.2 ± 3.4	27.0 ± 2.7	*0.163*
LBM (kg)	47.1 ± 8.4	49.5 ± 8.6	*0.502*
Body fat (%)	31.5 ± 9.9	35.0 ± 8.4	*0.363*
Systolic BP (mmHg)	144.6 ± 13.7	139.2 ± 18.8	*0.439*
Diastolic BP (mmHg)	88.5 ± 11.5	85.2 ± 10.4	*0.480*
HbA1c (mmol/L)	6.1 ± 0.4	6.0 ± 0.4	*0.702*
TSH (IU/L)	1.55 ± 0.67	1.73 ± 0.96	*0.605*
Total Cholesterol (mmol/L)	5.4 ± 0.7	5.6 ± 1.1	*0.557*
HDL (mmol/L)	1.65 ± 0.27	1.94 ± 0.49	*0.081*
LDL (mmol/L)	3.17 ± 0.71	3.19 ± 0.88	*0.940*

BMI: body mass index, LBM: lean body mass, BP: blood pressure, TSH: thyroid stimulating hormone, HDL: high density lipoprotein, LDL: low density lipoprotein. Data are shown as mean ± SD.

**Table 2 nutrients-14-04442-t002:** Meal distribution during the trial period.

**PROTEIN DISTRIBUTION**						
	**Breakfast**	**Snack**	**Lunch**	**Snack**	**Dinner**	*Total protein* g/kg LBM/day
EVEN (%)	30	5	30	5	30	*1.5*
SKEWED (%)	15	5	15	5	60	*1.5*
**ENERGY DISTRIBUTION**						
	**Breakfast**	**Snack**	**Lunch**	**Snack**		**Dinner**
EVEN (%)	30	5	30	5	30	
SKEWED (%)	30	5	30	5	30	

In both groups, the protein intake was 1.5 g/kg LBM/day. In the EVEN group, the dietary protein was distribution with 30% at breakfast, 5% as a noon snack, 30% at lunch, 5% as an afternoon snack and 30% at dinner. In the SKEWED group, protein distribution was 15% at breakfast, 5% as a noon snack, 15% at lunch, 5% as an afternoon snack and 60% at dinner. Both intervention groups had their daily calorie intake distributed evenly with 30% at breakfast, 5% as a noon snack, 30% at lunch, 5% as an afternoon snack and 30% at dinner.

**Table 3 nutrients-14-04442-t003:** Activity monitoring during habituation and trial days:.

	EVEN		SKEWED		*Two-way RM ANOVA*		
	Habituation	Trial	Habituation	Trial	*Time*	*Group*	*Interaction*
Steps	8923 ± 3947	13425 ± 2202	10573 ± 3567	14192 ± 2751	*<0.001*	*0.350*	*0.566*
Sit-lie time	1007 ± 137	1081 ± 59	1069 ± 49	1092 ± 67	*0.084*	*0.191*	*0.244*
Stand time	290 ± 71	129 ± 40	243 ± 43	116 ± 47	*<0.001*	*0.085*	*0.289*
Step time	115 ± 50	140 ± 26	129 ± 35	145 ± 28	*0.005*	*0.463*	*0.520*
MET	34.2 ± 1.6	33.6 ± 1.1	34.8 ± 1.5	33.9 ± 1.3	*0.006*	*0.370*	*0.497*

Activity levels of the EVEN and SKEWED group, during the habituation and trial period, respectively, expressed as daily steps, sit-lie time [minutes], stand time [minutes], step time [minutes] and MET [kcal/kg/hour]. Data are shown as mean ± SEM.

## Data Availability

The data presented in this study are available on request from the corresponding author. The data are not publicly available due to privacy restrictions.

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
