# Peer review of "Comparing Even with Skewed Dietary Protein Distribution Shows No Difference in Muscle Protein Synthesis or Amino Acid Utilization in Healthy Older Individuals: A Randomized Controlled Trial"

_nutrients, 2022, doi:10.3390/nu14214442_

Round 1

Reviewer 1 Report

This is a well written account of a well conducted study with clear results.

 My only comments relate to relatively minor issues of presentation of results and discussion of the methodology.  

aFor the D2O study the text states “The FSR is expressed in % x hour-1” line 295,  but the results, Fig 4a, are shown as FSR %/d

bFor the 2H5-phenylalanine incorporation there is no mention of the method in terms of whether any attempt was made to correct the uptake into muscle protein, shown as MPE% in fig 4B, with the enrichment of plasma 2H5-phenylalanine, which was measured (lines 283-284).

   The paper would be more informative in relation to the  stable isotope methodology if the measured values of the enrichments in the blood were reported and discussed in relation to the uptake into muscle protein.

Reviewer 2 Report

This study from Ehlig-Jensen et al. evaluated the potential effects of a skewed versus even dietary protein intake distribution in healthy older men and women. The research provide ample protein (~1.2 g per kg body mass) and apply stable isotope tracer methods to evaluate the dietary impact on muscle protein synthesis rates over several days. While this is an impactful topic, the novelty of the study is seemingly low considering prior work in the topic. Further explanation of the drawbacks of the current literature (highlighting potential novelty) is absent. The strengths of the study are its habituation period, full dietary and physical activity control, and application of deuterium oxide tracer to measure protein synthesis over several days (to incorporate multiple anabolic/catabolic factors in the measurement). 

A key drawback of the study is that the intrinsically labeled protein applied over several days does not only represent “dietary” protein as the authors have stated. This is because of tracer re-cycling where dietary-derived tracer is synthesized into organs/tissues followed by release of the tracer and re-utilization by muscle through synthesis (i.e., recycling). Therefore, measurement over days results in an unclear evaluation of what is directly incorporated into muscle from the diet and what has been re-utilized from different tissues. 

Minor comments:

Title/Abstract

-       Based on statistical approaches, it is more appropriate to state “no differences” than “even” or equal” effects of the treatments. Please revise the title and other statements of findings/conclusion.

-       Define the abbreviation D2O.

-       Include raw data in the abstract.

-       The conclusion is not clear, please revise since “a diet sufficient in protein” seems out of place.

Introduction

-       I recommend shortening the first paragraph to narrow the topic to dietary protein intake and muscle maintenance in the older population.

-       Line 57 – remove “the” in front of protein synthesis and protein breakdown

-       Line 65 “dose” should be “doses”

-       Line 67 – “suggest” should be “suggests”

-       Line 73-75 – provide evidence to support the claim that “protein intake could be below 0.40 g per kg BW”.

-       Line 76 – “resultantly” should be “consequently”

-       Line 77 “optimal” should be “optimally”

-       Line 88-90 – the sentence is not grammatically correct, please revise.

-       Line 97-98 – Add “vs a skewed protein intake” or something alike.

Methods

-       What was the rationale for the chosen “skewed” protein distribution (15,15,60% for breakfast lunch and dinner)? Please provide an explanation and references where necessary.

-       The protein was provided per LBM, but it was stated that prior evidence indicates that protein intake per meal should be greater than 0.40 g per kg BW. Please explain the difference in approach.

-       Line 168 - it’s stated “three trial days” but elsewhere it’s stated 4 trial days. Please revise or clarify.

-       Please state the percentage of D2O? What was the dosing protocol based upon? Add references where necessary.

-       Line 251 – “solvent” is not the correct word. Please replace with a more appropriate word.

-       Please add a statement explaining/clarifying what was measured in the separated muscle samples on the LCMSMS and GCIRMS. 

-       FSR was expressed as % per hour whereas D2O is administered over days. Therefore, % per day is more appropriate as the FSR unit.

-       Mean ± SD is more appropriate to display given that SD is a better representation of the distribution of the current data set. Please revise to display mean ±SD

Results

-       The protein intake is expressed per LBM, but the intake was seemingly targeted around the 0.4 g protein per BW per meal. Please also display the data relative to total body mass. For clarity, I suggest the authors comment mostly on the protein intake per total body weight throughout the manuscript and present values per lean body mass where relevant. This would make comparisons to prior literature easier to understand.

Discussion

-       State that this is the first study in healthy older men and women in an energy balanced diet as others have also applied similar methods, but in different populations and with restricted dietary intake.

-       Line 340 states no difference between monitored physical activity of SKEW and EVEN, which could have affected the integrated MPS measurement. It is not clear how the same amount of physical activity would have differently affected MPS. Please clarify.  

-       Line 395-401 please provide more detailed commentary on the differences in protein intake levels and sources of dietary protein applied in the present study compared to Kim et al. and Mamerow et al.

-       A main limitation of the incorporation of intrinsically labeled protein over several days is the recycling of labeled amino acids from other tissues and/or organs into the muscle. This is a very important limitation that should be evaluated and explained in the discussion.

-       Line 418 – Add “per meal” when discussing 0.4 g protein per kg body weight as being needed for maximal stimulation. 

-       In the conclusion, it is more appropriate to state “no differences” than “even” or “the same” effects of the treatments. Please revise.
